# Preliminary Evidence That Taping Does Not Optimize Joint Coupling of the Foot and Ankle Joints in Patients with Chronic Ankle Instability

**DOI:** 10.3390/ijerph18042029

**Published:** 2021-02-19

**Authors:** Charles Deltour, Bart Dingenen, Filip Staes, Kevin Deschamps, Giovanni A. Matricali

**Affiliations:** 1Department of Orthopaedics, University Hospitals Leuven, KU Leuven, 3000 Leuven, Belgium; 2Reval Rehabilitation Research Centre, Faculty of Rehabilitation Sciences, Hasselt University, 3590 Diepenbeek, Belgium; bart.dingenen@uhasselt.be; 3Department of Rehabilitation Sciences, KU Leuven, 3000 Leuven, Belgium; filip.staes@kuleuven.be (F.S.); kevin.deschamps@kuleuven.be (K.D.); 4Department of Development and Regeneration, KU Leuven, 3000 Leuven, Belgium; giovanni.matricali@uzleuven.be; 5Institute of Orthopaedic Research and Training, University Hospitals Leuven, KU Leuven, 3000 Leuven, Belgium; 6Department of Orthopaedics, Foot & Ankle Unit, University Hospitals Leuven, KU Leuven, 3000 Leuven, Belgium

**Keywords:** chronic ankle instability, multisegment kinematics, running, taping, joint coupling

## Abstract

Background: Foot–ankle motion is affected by chronic ankle instability (CAI) in terms of altered kinematics. This study focuses on multisegmental foot–ankle motion and joint coupling in barefoot and taped CAI patients during the three subphases of stance at running. Methods: Foot segmental motion data of 12 controls and 15 CAI participants during running with a heel strike pattern were collected through gait analysis. CAI participants performed running trials in three conditions: barefoot running, and running with high-dye and low-dye taping. Dependent variables were the range of motion (RoM) occurring at the different inter-segment angles as well as the cross-correlation coefficients between predetermined segments. Results: There were no significant RoM differences for barefoot running between CAI patients and controls. In taped conditions, the first two subphases only showed RoM changes at the midfoot without apparent RoM reduction compared to the barefoot CAI condition. In the last subphase there was limited RoM reduction at the mid- and rearfoot. Cross-correlation coefficients highlighted a tendency towards weaker joint coupling in the barefoot CAI condition compared to the controls. Joint coupling within the taped CAI conditions did not show optimization compared to the barefoot CAI condition. Conclusions: RoM was not significantly changed for barefoot running between CAI patients and controls. In taped conditions, there was no distinct tendency towards lower mean RoM values due to the mechanical restraints of taping. Joint coupling in CAI patients was not optimized by taping.

## 1. Introduction

Lateral ankle sprains (LAS) are one of the most frequently occurring musculoskeletal injuries [1,2,3], not only in athletes, but also in the general population [2,3,4]. Persons facing LAS have a great tendency for developing long-term complaints mostly caused by chronic ankle instability (CAI) [3]. CAI after LAS ranges from around 30% to 80%, depending on how the condition is defined [2,4,5,6]. According to the International Ankle Consortium, CAI is a chronic condition with three major aspects: (1) history of one significant ankle sprain, (2) ankle joint “giving way” and/or recurrent sprain and/or feelings of instability, and (3) poor self-reported functionality [4].

Many publications have shown that patients with CAI have different foot and lower limb kinematics when compared to non-symptomatic subjects [5,7,8,9,10,11,12,13]. These kinematic changes follow the same tendency in walking and running [12]. With respect to running, it has been reported that patients with CAI present less dorsiflexion and more inversion at the rearfoot [5,9,12,14]. Moreover, some studies observed a more internally rotated position of the rearfoot and a more adducted calcaneus [5,12]. These observations at the rearfoot suggest a loose packed position of the ankle, which may leave the ankle in a mechanically less advantageous position for load acceptance, consequently increasing injury risk [5,10,11,14]. In contrast, De Ridder et al., using a multisegment foot model for the first time, found an increased rearfoot eversion and unchanged dorsiflexion of the foot [10]. They reported a greater medial forefoot inversion [10], which was confirmed by Moisan et al., who also reported increased midfoot inversion and tibial external rotation [12].

Next to altered foot–ankle segmental motion, patients with CAI have also been associated with alterations in joint coupling, both in walking conditions and in running. Joint coupling is a measure for determining the degree of synchronization of motion between well-defined joint segments. Identifying changes in joint coupling enables gaining insight into movement disturbances and continued dysfunctionality [15]. It is assumed that disruption of physiological joint coupling reflects an unhealthy sensorimotor control system, which may be an underlying cause for repetitive soft tissue trauma [15,16]. In CAI subjects, joint coupling is less synchronized, especially during running [16].

Taping is a popular treatment in patients with CAI and helps preventing recurrent LAS [5,11,17,18]. It was suggested that taping could probably counteract the observed kinematic changes by restricting excessive ankle motion [5,11,17]. Next to motion control, it is also believed that taping improves proprioception through stimulation of cutaneous feedback [5]. Several authors observed significant changes of foot–ankle segmental motion in taped CAI subjects [5,11,18]. Taping was found to keep the ankle joint in a more neutral position, both in the frontal and sagittal plane, during walking and running. Generally, taping results in a more closed packed position of the ankle, which is important to prevent injury [5,11]. To our knowledge, the effect of taping on foot joint coupling has only been addressed by Herb et al., reporting a lower magnitude of joint coupling and less shank–rearfoot coupling variability in CAI and healthy subjects during taped conditions [17].

The overall changes in gait kinematics of patients with CAI reported in different publications largely agree but differ somehow in temporal observations. Some authors observed the kinematic changes during the complete stance phase and others only during certain periods of stance phase and/or swing phase. A possible explanation could be the difference in research setup (e.g., barefoot versus shod conditions). The shank–rearfoot coupling has been well studied in a rigid foot model but offers a rather incomplete evaluation with respect to the other joints of the foot [19]. Given the potential influence of taping on range of motion (RoM) and joint coupling, this study addressed multisegmental foot–ankle motion and joint coupling in barefoot-taped CAI subjects during different subphases of the stance phase of running. We first hypothesized that patients with CAI would demonstrate an increased sagittal and frontal plane RoM at the rear- and midfoot and a suboptimal shank–rearfoot joint coupling. We further hypothesized the RoM would be reduced by non-elastic taping and that joint coupling would be optimized in this patient population.

## 2. Materials and Methods

Twenty-seven recreationally active university students (defined by at least 1.5 h of cardiovascular activity per week) were recruited in this study and are identical to the study of Deschamps et al. [5]. Participants were categorized in two groups: the CAI group (6 men, 9 women) and the control group (5 men, 7 women). Inclusion criteria for the CAI group were (1) a history of at least one significant ankle sprain and (2) a history of the ankle joint giving way as defined by Delahunt et al. [20]. This self-reported ankle instability was objectivated with the Cumberland Ankle Instability Tool (CAIT) [21], a validated ankle instability-specific questionnaire using a cut-off benchmark of ≤24 to define CAI. Exclusion criteria for both groups were (1) age under 18 or above 30; (2) previous surgery or fracture in either lower extremity; (3) other lesions to the musculoskeletal structures of either lower extremity and/or back at the moment of testing that have an impact on joint integrity and function (except for CAI in the CAI group); (4) recent participation in a rehabilitation program; and (5) systemic, neurological, and orthopedic diseases. Our selection criteria generally matched those formulated by the IAC [4], except for exclusion criteria (1) and (4). All participants read and signed the informed consent. The local ethics committee of University Hospitals Leuven approved the study protocol (ML8745/S54821).

Running analysis was performed at the Clinical Motion Analysis Laboratory of the University Hospitals Leuven. A 10 m walkway was used with the following measurement devices: a 3D motion analysis system, a plantar pressure platform, and a force platform. A passive optoelectronic motion analysis system (Vicon Motion System Ltd., Oxford Metrics, Yarnton, Oxford, UK) with 10 T-10 cameras was used to track the kinematic data (100 Hz). In the middle of the walkway, a custom-made force plate was imbedded (Advanced Mechanical Technology Inc., Watertown, MA, USA), covered with a pressure plate (Footscan, dimensions 0.5 m × 0.4 m, 4096 sensors, 2.8 sensors per cm^2^, RSscan International, Olen, Belgium). Data from the force plate and pressure plate were sampled at 200 Hz.

Joint kinematics were estimated by first placing retroreflective markers according to the Rizzoli 3D multisegment foot model using double-sided tape [22]. Once fully instrumented, participants were instructed to run barefoot with a heel strike pattern at a constant speed of 3.3 m s^−1^ (±10%). Running speed was assessed by monitoring the velocity of the reflective marker on the sacrum. A heel strike pattern was imposed in all conditions to minimize the influence of striking pattern. The control group only performed barefoot running trials whereas the CAI group additionally performed trials with low-dye (LD) and high-dye (HD) taping until at least three valid trials were registered for each condition. The order of these test conditions was randomly assigned. Taping with non-elastic sports tape (38 mm, All Products BVBA, Aalst, Belgium) was applied by one investigator (BD) [5]. Subjects were in supine position with the feet in neutral position while taping. HD taping technique was performed according to MacDonald [23], whereas LD taping technique was performed according to Vicenzino et al. [24]. This protocol led to the determination of four different conditions in the current study: CAI barefoot (CAI_BF), CAI high-Dye (CAI_HD), CAI low-Dye (CAI_LD), and a control group.

The collected spatiotemporal variables were running speed, stride time, stance time, and swing time. Statistical analysis on demographical and spatiotemporal parameters was performed using Wilcoxon Test and ANOVA. Kinematic waveforms associated to the Instituto Ortopedico Rizzoli 3D multisegment foot model were computed using the Vicon Foot model Plug-in (Aurion Srl, Milano, Italy) using Nexus 1.8 software. This five-segment model defines the 3D rotations between the shank (Sha), calcaneus (Cal), midfoot (Mid), metatarsus (Met), and hallux as rigid segments. The sagittal plane angle of the first metatarsophalangeal joint (F2Ps) was also computed. Ground reaction forces collected during the running trials provided an objective basis for defining the stance phase of each running cycle.

Following data processing, temporal normalization of all waveforms for a complete gait cycle was performed with an in-house made Matlab software platform (Matlab2016a, The Mathworks, Natick, MA, USA). Subsequently, the stance phase of the running cycle was subdivided into three specific subphases: The first phase, called the peak impact phase (PIP), encompassed the time window from initial contact (0%) to 5% of the running cycle. The second phase, called the absorption phase, included the time window from 6% of the running cycle to the peak vertical ground reaction force (GRF). The last phase went from the aforementioned peak GRF to toe-off (generation phase (GP)). Swing Phase (32–100%) was not included in our research. Range of motion, defined as the difference between the maximum and minimum value in each subphase, was first calculated for each trial and subsequently a mean value was calculated for the respective subject and this for each condition. Testing for differences between the control group and CAI_BF was performed using a two-sample *t*-test and comparison between the different CAI conditions was performed using a one-way repeated measures ANOVA. In order to retain a prescribed family-wise error rate alpha in the current statistical analysis, the error rate for each comparison was set at 0.5/3 = 0.017.

The level of kinematic coupling during the stance phase of the running cycle was evaluated by calculating the cross-correlation coefficient between the following four inter-segmental angles [19,25,26]: (1) Sha-Cal Inversion/Eversion with Sha-Cal Adduction/Abduction, (2) Sha-Cal Inversion/Eversion with Cal-Met Dorsiflexion/Plantarflexion, (3) Sha- Cal Inversion/Eversion with Cal-Met Inversion/Eversion, and (4) Sha-Cal Inversion/Eversion with Cal-Met Adduction/Abduction. The coupling of these specific segments represents the relationship between the shank, and rear- and forefoot [19,26]. The following qualitative benchmarks were applied when evaluating these cross-correlation coefficients: (1) strong coupling >0.7 or <−0.7, (2) moderate coupling between (−)0.3 to (−)0.69, and (3) weak coupling between −0.3 and 0.3 [26].

## 3. Results

### 3.1. Demographic and Spatiotemporal Data

Demographic data showed no significant differences among both groups and conditions (*p* > 0.05). There were some significant differences in spatio-temporal parameters between the control group and the CAI_BF condition. The demographic and spatiotemporal data are summarized in Table 1.

### 3.2. Comparison CAI_BF versus Controls

The mean RoM values during the three stance subphases of running in barefoot CAI subjects versus controls are presented in Table 2. No statistically significant differences were observed between these two groups.

### 3.3. Comparison CAI_BF versus CAI_HD versus CAI_LD

An overview of the mean RoM values associated to the different CAI conditions is provided in Table 3, Table 4 and Table 5. During the PIP and AP, there were only significant RoM differences in the Cal-Mid segment (Table 3 and Table 4). In the GP, there were significant RoM differences in three segments: Sha-Cal, Cal-Mid, and Mid-Met (Table 5).

### 3.4. Joint Coupling

Correlational analyses performed to assess the degree of joint coupling throughout the different conditions revealed a tendency towards a weaker joint coupling in the barefoot CAI condition compared to the control subjects, except for the Sha-Cal Inv/Eve Cal-Met Add/Abd coupling. When the CAI conditions are compared to each other, the same tendency is noticed (Table 6).

## 4. Discussion

In this study, motion of well-defined foot segments during distinct subphases of the stance phase of running was compared between a control group and three CAI conditions. The mean RoM of the controls and CAI_BF showed no significant differences; therefore, clear differences were observed among the CAI conditions. Ultimately, evidence for suboptimal joint coupling has been illustrated among the CAI group, even after the application of non-elastic taping techniques.

Based on the current findings, we have to reject our first hypothesis, that is, that patients with CAI demonstrate increased sagittal and frontal plane RoM at the rear- and midfoot during specific subphases of the stance phase of running. However, previous studies have shown that the absolute segmental positions were affected by CAI, represented by a less dorsiflexed, more internally rotated, and inverted rearfoot together with a more inverted mid- and forefoot [5,9,12,14]. Our findings can be explained by the forced heel strike pattern, which constrains the foot–ankle complex during the rest of the stance phase. This mechanism probably avoids an increased loose packed ankle joint position observed in other studies [5,6,10,11,14]. Striking pattern is an important factor in the analysis of the stance phase of the running cycle as indicated by a study from Deschamps et al. They reported that running with a midfoot striking pattern provided a decreased sagittal rearfoot RoM during the PIP, an increased transverse and sagittal rearfoot plane RoM during the AP, and an increased frontal midfoot plane RoM during the GP [6]. In contrast with our findings, this suggested a change in neuromuscular control and risk factor for recurrent LAS [6].

In taped conditions, there were several significant changes compared to the CAI_BF condition, however there was no clear pattern. We hypothesized a decreased mean RoM due to the mechanical restrictions of taping, but our results did not show this consistently. In the PIP, there were significant RoM changes in the sagittal and frontal plane at the Cal-Mid segment. HD taping showed a larger RoM in the sagittal plane in this inter-segmental angle compared to the other two CAI conditions. A closer look at the kinematic waveforms of Deschamps et al. highlights that this finding mainly represents an increased plantarflexion [5]. This is probably due to the restraint of the HD taping at the ankle joint and the forced heel strike, which aids in locking the ankle position. These two factors force the Cal-Mid segment to compensate towards plantarflexion. The CAI_LD condition only had a lower frontal plane RoM at the Cal-Mid segment than the barefoot condition, representing a reduced eversion [5]. As Sha-Cal RoM in the PIP was found to be similar among the taping conditions, it is believed that this may originate from subtle differences in foot loading during this subphase, especially with a forced heel strike pattern.

In the AP, we observed a significant lower RoM in the sagittal plane at the Cal-Mid angle with LD taping compared to the CAI_BF condition and HD taping condition. This corresponds to reduced dorsiflexion when kinematic waveforms are assessed [5]. Significantly decreased sagittal plane RoM in the HD taping condition at the Sha-Cal segment is seen in the GP, which signifies reduced plantarflexion [5]. In the CAI_LD condition, this is not observed as the LD taping does not include the tibiotalar joint, where most of the sagittal plane motion of the Sha-Cal segment occurs. The RoM reduction at the Sha-Cal segment during GP suggests that HD taping considerably affects the power generation capacity of the ankle joint, which may be an interesting finding from a sports performance perspective and the unraveling of compensatory strategies. The most obvious strategy is compensation by adjacent joints in the kinetic chain, most likely the mid- and forefoot. However, our results do not demonstrate this. An alternative strategy for generating more propulsive power could be a shift to more proximal joints such as the knee and hip.

The significant loss of RoM at the Cal-Mid and Mid-Met segments are more pronounced in the CAI_LD condition than in the CAI_HD condition. Kinematic waveforms of these segments during GP show a decreased inversion and adduction at the Cal-Mid segment, and a decreased plantarflexion and abduction at the Mid-Met segment [5]. Next to the mechanical restrictions that the non-elastic taping is providing (passive support of the medial arch), it is also plausible that these observed kinematic effects of the LD taping originate from stimulation of the plantar mechanoreceptors, located in the medial and lateral aspect of the midfoot [27,28]. Indeed, preliminary evidence suggests that stimulation of these mechanoreceptors may enhance muscle activity of both intrinsic and extrinsic foot muscles [27,28]. Moreover, HD taping covers a larger area and includes the ankle joint itself, which suggests a different proprioceptive stimulation compared to LD taping. The decreased RoM in the midfoot and forefoot in presence of LD taping may also affect the load distribution at the plantar surface of the foot as there is preliminary evidence about the correlation between segmental foot motion and this load distribution [29].

Several authors have recently investigated the effect of elastic taping (e.g., Kinesiotape) on CAI. It is assumed that elastic taping has less mechanical restraint and exercises its effect more through sensorimotor control [30,31,32]. Moreover, it can generate a pulling force that delivers a sensory cue which guides correct ankle motion [30]. In contrast to our research, the effect of elastic taping has been investigated in walking, landing tasks, and postural control without consideration of joint coupling in a running multisegmental foot model. Despite these differences in research set-up, a comparison can be useful. Yen et al. reported that elastic taping facilitates foot eversion during early stance without restricting natural eversion during late stance in shod walking CAI subjects [30]. In the non-elastic taping condition, they found no influence on RoM, based on the assumption that mechanical restraint of the taping does not happen in the functional RoM [30]. This finding supports our results that RoM in barefoot running CAI subjects is barely affected by non-elastic taping. When landing tasks are considered, Kuni et al. found a more stabilizing effect of non-elastic taping at the rear- and midfoot, compared to elastic taping [32]. De Ridder et al. discovered that even limited non-elastic taping in barefoot CAI subjects reduced inversion and plantarflexion of the foot during landing tasks [33]. In postural control tests, no significant influence of elastic and non-elastic taping was observed. However, both elastic and non-elastic taping offered a greater perceived stability [34].

Next to multisegmental RoM analysis of different CAI conditions, we also examined the degree of joint coupling. To our knowledge, this is the first study to evaluate multisegmental joint coupling in taped CAI subjects. Pohl et al. studied multisegmental ankle-foot joint coupling in normal running subjects and discovered a strong coupling between rearfoot (Sha-Cal) frontal plane (Inv/Eve) and transverse plane motion (Add/Abd) [19]. The rearfoot Inv/Eve was also strongly coupled with forefoot (Cal-Met) sagittal plane (DF/PF) and transverse plane (Add/Abd) motion [19]. The only weak coupling of the rearfoot was with forefoot frontal (Inv/Eve) motion [19]. This strong coupling was not reported in our control subjects. The study population of Pohl et al. had a natural heel strike pattern opposed to our forced heel strike. This possibly weakened the joint coupling in our controls. Another study setup and slightly different segment definition can contribute to these findings. We hypothesized that CAI would lead to a weaker joint coupling compared to controls, as described in other publications [15,16]. In the taped conditions, we expected an optimization of joint coupling. Based on our results, we cannot maintain these hypotheses. There is a trend towards a weaker joint coupling in all CAI conditions. At the Sha-Cal Inv/Eve Sha-Cal Add/Abd coupling HD taping caused a negligible optimization compared to CAI_BF. However, LD taping showed a slightly better optimization of this coupling. The same trend is observed at the Sha-Cal Inv/Eve Cal-Met Inv/Eve coupling where LD coupling has a tendency towards more optimization of coupling than HD taping. These findings were not expected, as LD taping only promotes passive stability of the midfoot and potentially stimulation of the midfoot plantar mechanoreceptors, whereas in HD taping these potential actions are extended towards stabilization of proximal joints as well as proprioceptive stimulation around the ankle region. Possibly HD taping disables the coupling between shank and calcaneus because of its mechanical restraints, as suggested by others [5,17]. Using vector coding, Herb et al. detected a lower magnitude of joint coupling in taped controls and CAI subjects only during swing phase while running [17]. This was attributed to the mechanical restrictions of the tape application. Moreover, a decreased joint coupling variability was also observed, which led to the conclusion that this more consistent gait pattern offers a protection against recurrent LAS [17]. It can be stated that in our results the LD taping resulted in slightly better kinematic properties than HD taping. The heel strike landing is probably an important factor in this observation. Stating that a heel strike landing combined with LD taping is a possible coping strategy for CAI, should, however, be done cautiously. It is plausible that this can be helpful in patients that already have a natural heel strike pattern during running.

Although new insights are provided, especially concerning multisegmental joint coupling in taped CAI subjects, there are several limitations to this study. First, the sample size was rather small in the current study. Second, analysis focused on barefoot running, which may not reflect realistic conditions. Therefore, extrapolation of our results to shod conditions should be done cautiously. However, unfortunately multisegmental foot kinematics are difficult to examine in shod conditions. Third, participants were forced to adopt a heel strike pattern, probably interfering with the natural landing strategy of some participants. Consequentially, the full biomechanical picture can be obscured. Future challenges to further improve the external validity of the research in this field could be a larger sample size in which participants use their natural landing strategy with analysis of barefoot and shod running. Furthermore, the control subjects can also be analyzed with taping to obtain a complete picture of taping effects on multisegmental foot kinematics.

## 5. Conclusions

The multisegmental kinematic research in this study was unable to demonstrate significant RoM differences between the CAI_BF condition and controls during the stance phase of running with a heel strike pattern. Moreover, in taped conditions there was no distinct tendency towards lower mean RoM values due to the mechanical restraints of taping compared to CAI_BF. The observed RoM differences in taped conditions were slightly more pronounced with LD taping, probably attributed to a different proprioceptive stimulation. In all CAI conditions, there was a trend towards a weaker joint coupling, despite application of non-elastic taping. Further research is needed to confirm our findings and possibly establish guidelines for adequate therapeutic interventions in patients with CAI.

## Figures and Tables

**Table 1 ijerph-18-02029-t001:** Demographic and spatio-temporal data of both groups.

Demographic Characteristics	Control	CAI		
Subjects	12	15		
Age (years)	23.6 ± 4.1	22 ± 2.7		
BMI (kg/m^2^)	22.2 ± 2.0	23.6 ± 3.0		
Male/Female	4/7	6/9		
**Spatio-temporal parameters**	**Control**	**CAI_BF**	**CAI_LD**	**CAI-HD**
Running speed (m/s)	3.6 ± 0.3 *	3.2 ± 0.3 *	3.2 ± 0.4	3.2 ± 0.4
Stride time (s)	0.7 ± 0.1	0.7 ± 0.01	0.8 ± 0.1	0.7 ± 0.1
Swing time (% RC)	69.1 ± 1.8 *	65.8 ± 2.1 *	67.1 ± 2.6	67.4 ± 2.5
Stance time (% RC)	30.9 ± 1.8 *	34.2 ± 2.1 *	32.9 ± 2.6	32.6 ± 2.5

BMI = body mass index, RC = running cycle, CAI_BF: Chronic ankle instability group, barefoot condition; CAI_HD: Chronic ankle instability group, high-dye taping condition; CAI_LD: Chronic ankle instability group, low-dye taping condition. * Groups significantly different at *p* < 0.05 level.

**Table 2 ijerph-18-02029-t002:** Summary of mean range of motion and standard deviation (in degrees) during three subphases of stance while running in controls versus CAI_BF.

		Peak Impact Phase (0–5%)			Absorption Phase (6–16%)			Generation Phase (17–31%)		
		Controls	CAI_BF	*p* Value	Controls	CAI_BF	*p* Value	Controls	CAI_BF	*p* Value
Sha-Cal	DF/PF	3.6 (1.9)	3.5 (1.2)	0.8026	11.1 (3.1)	10.1 (2.7)	0.8808	37.4 (5.2)	38.6 (3.9)	0.9045
	Inv/Eve	2.9 (2.0)	3.2 (1.7)	0.8026	2.5 (1.0)	2.2 (1.6)	0.6171	5.5 (2.9)	4.1 (2.0)	0.5892
	Add/Abd	1.6 (1.1)	1.4 (1.2)	0.9203	3.4 (1.9)	3.6 (2.2)	0.9840	7.4 (3.8)	8.4 (3.7)	0.9840
Cal-Mid	DF/PF	2.0 (1.9)	1.6 (1.1)	0.7039	6.9 (1.5)	6.0 (2.2)	0.5961	15.2 (3.9)	15.5 (5.2)	0.9283
	Inv/Eve	4.8 (2.0)	4.9 (1.6)	0.9840	2.3 (1.1)	3.2 (1.5)	0.5029	5.2 (1.1)	7.3 (2.5)	0.4065
	Add/Abd	0.9 (0.6)	1.7 (1.6)	0.7188	1.4 (0.8)	1.7 (0.6)	0.8259	6.8 (1.7)	6.5 (2.7)	0.4593
Mid-Met	DF/PF	1.2 (0.9)	1.6 (1.1)	0.9124	1.7 (0.8)	1.8 (1.4)	0.6891	5.2 (1.5)	4.7 (2.0)	0.7263
	Inv/Eve	1.6 (1.3)	2.3 (2.0)	0.8572	1.3 (0.8)	1.5 (0.9)	0.7949	3.4 (1.5)	3.2 (1.4)	0.8650
	Add/Abd	2.6 (2.0)	1.9 (1.9)	0.9045	1.4 (0.7)	1.9 (1.2)	0.4533	5.0 (1.6)	5.2 (1.6)	0.7414
Cal-Met	DF/PF	2.2 (1.7)	1.9 (1.1)	0.9124	6.9 (1.7)	6.9 (2.0)	0.9203	19.2 (4.5)	19.3 (3.8)	0.7795
	Inv/Eve	1.6 (1.0)	2.5 (1.2)	0.2340	1.3 (1.1)	1.6 (1.3)	0.8650	6.6 (1.7)	5.7 (2.8)	0.7718
	Add/Abd	2.9 (2.3)	4.3 (1.7)	0.6599	1.8 (0.8)	2.6 (1.6)	0.5353	5.7 (1.7)	5.9 (2.3)	0.7279
F2Ps	DF/PF	7.6 (3.0)	7.3 (3.9)	0.8493	8.3 (4.0)	7.5 (3.7)	0.5823	30.4 (4.1)	29.2 (5.3)	0.4009

DF/PF = Dorsiflexion/Plantarflexion, Inv/Eve = Inversion/Eversion, Add/Abd = Adduction/Abduction. Sha-Cal = Shank-Calcaneus, Cal-Mid = Calcaneus-Midfoot, Mid-Met = Midfoot-Metatarsus, Cal-Met = Calcaneus-Metatarsus, F2Ps = 1st Metatarsophalangeal Joint.

**Table 3 ijerph-18-02029-t003:** Summary of mean RoM and standard deviation (in degrees) during peak impact phase in CAI_BF versus CAI_HD versus CAI_LD.

		Peak Impact Phase (0–5%)			
		CAI_BF ^1^	CAI_HD ^2^	CAI_LD ^3^	*p* Value
Sha-Cal	DF/PF	3.5 (1.2)	2.8 (1.4)	3.9 (2.6)	0.3098
	Inv/Eve	3.2 (1.7)	3.2 (1.7)	2.4 (1.4)	0.0192
	Add/Abd	1.4 (1.2)	1.8 (2.1)	2.9 (3.5)	0.0649
Cal-Mid	DF/PF	1.6 (1.1) *^,2^	4.3 (2.6) *^,1,3^	1.6 (1.1) *^,2^	0.0089
	Inv/Eve	4.9 (1.6) *^,3^	3.3 (1.3)	2.8 (1.1) *^,1^	0.0065
	Add/Abd	1.7 (1.6)	1.1 (0.9)	0.9 (0.6)	0.6835
Mid-Met	DF/PF	1.6 (1.1)	2.5 (2.0)	1.6 (1.2)	0.2700
	Inv/Eve	2.3 (2.0)	1.9 (1.2)	1.4 (1.3)	0.2583
	Add/Abd	1.9 (1.9)	1.1 (1.1)	1.8 (1.3)	0.3476
Cal-Met	DF/PF	1.9 (1.1)	2.7 (1.2)	2.0 (1.1)	0.1203
	Inv/Eve	2.5 (1.2)	1.9 (1.0)	1.5 (0.8)	0.0772
	Add/Abd	4.3 (1.7)	3.4 (1.4)	2.7 (1.2)	0.0220
F2Ps	DF/PF	7.3 (3.9)	9.6 (5.7)	7.4 (4.5)	0.5009

DF/PF = Dorsiflexion/Plantarflexion, Inv/Eve = Inversion/Eversion, Add/Abd = Adduction/Abduction. Sha-Cal = Shank-Calcaneus, Cal-Mid = Calcaneus-Midfoot, Mid-Met = Midfoot-Metatarsus, Cal-Met = Calcaneus-Metatarsus, F2Ps = 1st Metatarsophalangeal Joint. * Indicates significant differences between groups (groups are coded with a number: CAI_BF = ^1^, CAI_HD = ^2^ and CAI_LD = ^3^).

**Table 4 ijerph-18-02029-t004:** Summary of mean RoM and standard deviation (in degrees) during absorption phase in CAI_BF versus CAI_HD versus CAI_LD.

		Absorption Phase (6–16%)			
		CAI_BF ^1^	CAI_HD ^2^	CAI_LD ^3^	*p* Value
Sha-Cal	DF/PF	10.1 (2.7)	10.4 (3.1)	12.3 (3.3)	0.2764
	Inv/Eve	2.2 (1.6)	2.0 (0.8)	2.7 (1.1)	0.3625
	Add/Abd	3.6 (2.2)	3.1 (2.1)	3.7 (2.0)	0.7558
Cal-Mid	DF/PF	6.0 (2.2) *^,3^	6.8 (1.4) *^,3^	4.3 (1.1) *^,1,2^	0.0015
	Inv/Eve	3.2 (1.5)	2.9 (1.5)	2.3 (0.7)	0.2495
	Add/Abd	1.7 (0.6)	1.4 (0.9)	1.1 (0.6)	0.0638
Mid-Met	DF/PF	1.8 (1.4)	1.8 (1.0)	2.1 (1.7)	0.8345
	Inv/Eve	1.5 (0.9)	1.7 (1.1)	1.4 (0.6)	0.7225
	Add/Abd	1.9 (1.2)	1.1 (0.6)	1.2 (0.8)	0.2870
Cal-Met	DF/PF	6.9 (2.0)	6.5 (1.1)	5.3 (2.5)	0.1793
	Inv/Eve	1.6 (1.3)	1.7 (1.2)	1.5 (1.1)	0.7888
	Add/Abd	2.6 (1.6)	2.8 (1.7)	2.1 (1.4)	0.6055
F2Ps	DF/PF	7.5 (3.7)	6.4 (2.9)	10.2 (8.4)	0.5899

DF/PF = Dorsiflexion/Plantarflexion, Inv/Eve = Inversion/Eversion, Add/Abd = Adduction/Abduction. Sha-Cal = Shank-Calcaneus, Cal-Mid = Calcaneus-Midfoot, Mid-Met = Midfoot-Metatarsus, Cal-Met = Calcaneus-Metatarsus, F2Ps = 1st Metatarsophalangeal Joint. * Indicates significant differences between groups (groups are coded with a number: CAI_BF = ^1^, CAI_HD = ^2^ and CAI_LD = ^3^).

**Table 5 ijerph-18-02029-t005:** Summary of mean RoM and standard deviation (in degrees) during generation phase in CAI_BF versus CAI_HD versus CAI_LD.

		Generation Phase (17–31%)			
		CAI_BF ^1^	CAI_HD ^2^	CAI_LD ^3^	*p* Value
Sha-Cal	DF/PF	38.6 (3.9) *^,2^	32.1 (4.3) *^,1,3^	43.2 (3.8) *^,2^	0.0000
	Inv/Eve	4.1 (2.0)	4.6 (2.8)	4.3 (2.6)	0.8426
	Add/Abd	8.4 (3.7)	7.8 (4.4)	10.5 (3.9)	0.3570
Cal-Mid	DF/PF	15.5 (5.2)	17.9 (4.5)	16.6 (4.6)	0.5873
	Inv/Eve	7.3 (2.5) *^,3^	6.5 (1.2) *^,3^	4.6 (1.1) *^,1,2^	0.0011
	Add/Abd	6.5 (2.7) *^,2,3^	3.0 (1.4) *^,1^	3.5 (2.0) *^,1^	0.0010
Mid-Met	DF/PF	4.7 (2.0) *^,3^	3.0 (1.8)	2.4 (1.6) *^,1^	0.0056
	Inv/Eve	3.2 (1.4)	2.3 (1.3)	2.4 (1.1)	0.2167
	Add/Abd	5.2 (1.6) *^,2,3^	3.5 (1.4) *^,1^	3.4 (1.6) *^,1^	0.0014
Cal-Met	DF/PF	19.3 (3.8)	20.5 (3.4)	16.9 (4.4)	0.0963
	Inv/Eve	5.7 (2.8)	3.6 (1.5)	4.4 (3.0)	0.1094
	Add/Abd	5.9 (2.3)	5.8 (1.3)	5.1 (1.9)	0.6667
F2Ps	DF/PF	29.2 (5.3)	23.0 (6.0)	31.2 (10.1)	0.0380

DF/PF = Dorsiflexion/Plantarflexion, Inv/Eve = Inversion/Eversion, Add/Abd = Adduction/Abduction. Sha-Cal = Shank-Calcaneus, Cal-Mid = Calcaneus-Midfoot, Mid-Met = Midfoot-Metatarsus, Cal-Met = Calcaneus-Metatarsus, F2Ps = 1st Metatarsophalangeal Joint. * Indicates significant differences between groups (groups are coded with a number: CAI_BF = ^1^, CAI_HD = ^2^ and CAI_LD = ^3^).

**Table 6 ijerph-18-02029-t006:** Mean cross-correlations for the four participant groups.

	Control	CAI_BF	CAI_HD	CAI_LD
Sha-Cal Inv/Eve Sha-Cal Add/Abd	0.68	0.45	0.50	0.63
Sha-Cal Inv/Eve Cal-Met DF/PF	−0.82	−0.46	−0.57	−0.49
Sha-Cal Inv/Eve Cal-Met Inv/Eve	−0.45	−0.19	0.00	−0.34
Sha-Cal Inv/Eve Cal-Met Add/Abd	0.43	0.58	0.60	0.55

DF/PF = Dorsiflexion/Plantarflexion, Inv/Eve = Inversion/Eversion, Add/Abd = Adduction/Abduction. Sha-Cal = Shank-Calcaneus, Cal-Met = Calcaneus-Metatarsus.

## Data Availability

The data presented in this study are available on request from the corresponding author.

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
