# Peer review of "Preliminary Evidence That Taping Does Not Optimize Joint Coupling of the Foot and Ankle Joints in Patients with Chronic Ankle Instability"

_ijerph, 2021, doi:10.3390/ijerph18042029_

Round 1

Reviewer 1 Report

CAI is a really important topic, given prevalence and chronicity, morbidity.

Taping is a frequent intervention, and more knowledge of effect is helpful.

So, the topic and orientation of this study is good, but the design has greatly limited the external validity of this study, viz.

  • small and age-limited sample
  • only barefoot running, no use of shoes - usual or standardised!
  • limited use of control comparison, ie only barefoot - would be helpful to better understand tape use if also included control comparison eg Line 240

    'In taped conditions there were several significant changes compared to the barefoot condition,  however there was no clear pattern.' - but there is no control group comparison possible across the 3 conditions

  • participants 'forced' to adopt heel strike vs usual running pattern; another likely confounder, and curious as to why, when sample so small?

Lesser points:

  • title is premature (evidence?, perhaps 'preliminary trend')
  • definition of 'joint coupling' needs to appear in introduction
  • comparison with Pohl study is a stretch, given different methodologies

I suggest that:

  • this study design is improved, ie shoes, natural running style (can sub group HS vs pose etc)
  • complete control group comparisons
  • larger sample with greater demographic variation if possible (or comparison with another distinct cohort of eg, older runners).

A great topic, but the design needs considerable improvements, which if possible as outlined, may make this investigation suitable to reconsider.

Reviewer 2 Report

Congratulation for the wonderful paper. Thank you for this helpful contribution. I applaud the effort of promoting studies for the investigation of non-invasive procedures. I appreciate their methods including study design and data analysis. I write some comments below that could benefit the article.

Results. Tables. Information is shown in duplicate. If it appears in the table, delete the information from the text.

Conclusion. Conclusion must respond to the main aim of the study. It is advisable that conclusion be clear and concise. Therefore, it is not recommended that its length be greater than some lines.

References. I would like to congratulate you because 100% of the information you have referenced comes from articles published in scientific journals.

Reviewer 3 Report

Dear Authors,

congratulations on this paper. I find it very interesting and useful.

The theme is interesting for people dealing with treatment of chronic ankle instability. This study supports the opinion that taping of unstable ankle joint does not influence on gait and foot control during walking. But there are other literature data that support opposite opinion.  I would like to read discussion on these topics.

The idea is not original at all, but it is well presented and phases of gait are well analysed and compared. Recent published material should be better discussed.

I did not see You mentioned and commented some recent literature data. Just few of them are;

  • doi: 10.1123/jsr.2018-0234
  • doi: 10.1016/j.gaitpost.2018.08.034
  • doi: 10.1123/jsr.2019-003

Best regards,

Reviewer

Reviewer 4 Report

Purpose of study, review of literature and methodology design are appropiate without any careless mistakes. Finding of the study can answer the reserach questions and make significant contributions to the rehabilitative area.

The results of this study will be helpful for the individual with CAI to know more about the effect of taping in coupling of foot and ankle join in running. The literature of taping with CAI, importance of joint coupling are well addressed at the very beginning. The uniqueness of study is well defined and the research gap is clearly stated. The conclusions are drawn according to the results of the study which are also aligned with the hypotheses and answered the research questions.

Round 2

Reviewer 1 Report

Thank you to the authors for improving this manuscript, and it is now a much more balanced report. The reporting of limitations is far better, and the preliminary nature of these results are better, clearer and I hope, a better baseline for future studies investigating CAI.